# Association of Body Mass Index, Blood Pressure, and Interictal Serum Levels of Cytokines in Migraine with and without Aura

**DOI:** 10.3390/jcm11195696

**Published:** 2022-09-26

**Authors:** Aelita Plinta, Peteris Tretjakovs, Simons Svirskis, Inara Logina, Gita Gersone, Antra Jurka, Indra Mikelsone, Leons Blumfelds, Vitolds Mackevics, Guntis Bahs

**Affiliations:** Faculty of Medicine, Riga Stradins University, 16 Dzirciema Str., LV-1007 Riga, Latvia

**Keywords:** migraine, aura, cytokines, body mass index, blood pressure

## Abstract

The aim of the study was to clarify correlations between body mass index (BMI), blood pressure (BP), and serum levels of cytokines in female migraine patients. A total of 14 migraineurs with aura, and 12 without aura during their interictal period were compared with 25 controls. Interleukin-8 (IL-8), soluble intercellular adhesion molecule-1 (sICAM-1), soluble vascular cell adhesion molecule-1 (sVCAM-1), matrix metalloproteinase-9 (MMP-9), interferon gamma (IFN-γ), monocyte chemoattractant protein-1 (MCP-1), transforming growth factor alpha (TGF-α), and plasminogen activator inhibitor-1 (PAI-1) were measured. Migraineurs have elevated levels of IL-8, but decreased serum levels of PAI-1 and sICAM-1 during the interictal period, regardless of aura. BMI correlates with BP, and also with IFN-γ and MMP-9 only in patients with aura. There are three correlations in migraine patients with aura that are absent in patients without aura: between IL-8 and PAI-1; MMP-9 and IL-8; and IL-8 and sICAM-1. Migraineurs without aura, on the other hand, have correlations that patients with aura do not have (between PAI-1 and MCP-1, sICAM-1; between MMP-9 and sICAM-1, MCP-1; between TGF-α and PAI-1, MMP-9, sICAM-1; between sICAM-1 and MMP-9, PAI-1, MCP-1; as well as between sVCAM-1 and MCP-1). PAI-1, TGF, and MMP-9 could be used as biomarkers to distinguish migraineurs from healthy individuals.

## 1. Introduction

About 11% of the world’s population suffers from migraines [1]. Migraine, a chronic neurovascular disorder, is characterized by episodic headaches and has a multifaceted pathogenesis. The pathogenesis is associated with a relative decrease in cerebral circulation and subsequent reactive hyperemia, as well as sterile inflammation and hypersensitivity to pain [2]. According to the trigeminal neurovascular reflex theory, the migraine pain is related to the release of calcitonin gene-related peptide (CGRP) from the perivascular nerves due to trigeminal activation, and CGRP causes vasodilatation and neurogenic inflammation [3]. Studies suggest that cytokines, e.g., interleukin (IL)-1β, tumour necrosis factor alpha, IL-10, and IL-4, may play a role in pain modulation and the development of trigeminal nerve fibers sensitization [3,4]. Cytokines can act on neuronal receptors, cause neurovascular inflammation, and contribute to pain [5].

Being overweight, and obesity in particular, is associated with a higher risk of developing not only diabetes, hypertension, cardiovascular disease, and other diseases, but also a variety of pain disorders. The results of the meta-analysis show that there is a link between migraine and obesity [6]. Data suggest that migraineurs with a higher body mass index (BMI) suffer from a higher incidence, severity, and duration of headaches [1].

In general, hypertension plays one of the main roles in cardiometabolic diseases. Regarding the association between migraine and hypertension or hypotension, published data are contradictory [7], although migraines are shown to be a risk factor for the development of hypertension [8]. It should be noted that one study found an inverse relationship between blood pressure (BP) and the development of headaches [9]. Overall, studies confirm the link between migraines and BP, but it is complex and multifactorial [10].

We hypothesize that there are associations between BMI, BP, and inflammatory cytokines in migraine patients. That is why we set the aim of the study as clarifying correlations among BMI, BP, and serum levels of cytokines in female migraine patients in an interictal period.

## 2. Materials and Methods

### 2.1. Study Subjects

Twenty-six migraine patients (without aura 12, with aura 14) during their interictal period were compared with twenty-five healthy control subjects [11]. Only women, not in the menstrual period, were included in the study. Both groups were matched with regard to their age, body mass index, and blood pressure. Healthy subjects were excluded if they had first-degree relatives suffering from migraine. The baseline characteristics for both study groups are shown in Table 1.

Exclusion criteria were: history of cardiovascular disease, arterial hypertension, hyperlipidemia, diabetes mellitus, and also smoking. Other exclusion factors were thyroid dysfunction, acute or chronic inflammatory state, renal or liver diseases, malignancies, and other diseases that are known to be associated with significant changes of cytokines. No regular medication was allowed, including hormonal contraceptive method.

This study was carried out in accordance with the Declaration of Helsinki. All subjects gave their informed consent to the protocol, which was approved by the Medical Ethics Committee of the Riga Stradins University for biomedical research (No. 24.02.2012/1).

### 2.2. Laboratory Assays

The venous blood samples were collected from the study subjects after their overnight fasting, they were centrifuged, and stored at −80 °C. Interleukin-8 (IL-8)–Merck Millipore, EZHIL8-100K; soluble intercellular adhesion molecule-1 (sICAM-1)–Merck Millipore, ECM335; soluble vascular cell adhesion molecule-1 (sVCAM-1)–Merck Millipore, ECM340; matrix metalloproteinase-9 (MMP-9)–Sigma-Aldrich, RAB0372-1KT; interferon gamma (IFN-γ)–Sigma-Aldrich, RAB0222-1KT; monocyte chemoattractant protein-1 (MCP-1)–Merck Millipore, EZMCP1-99KRM; and transforming growth factor alpha (TGF-α)–Sigma-Aldrich, RAB0459-1KT were measured in serum by ELISA method using TECAN Infinite 200 PRO multimode reader (Tecan Group, Ltd., Mannedorf, Switzerland). Concentrations of lipids, glucose, and other routine blood biomarkers were analyzed by standard methods.

### 2.3. Statistical Analysis

Normal distribution of data was proven by D’Agostino and Pearson, Anderson–Darling, and Shapiro–Wilk normality tests. Homogeneity of variances was tested using F-test (2 groups) or Brown–Forsythe and Bartlett’s tests (≥3 groups). Between-group comparisons were performed using unpaired t-test or ordinary one-way ANOVA. Where data were non-normally distributed, non-parametric Mann–Whitney (MW) U-test or Kruskal–Wallis (KW) H-test, followed by two-stage step-up method of Benjamini, Krieger, and Yekutieli as post-hoc procedure, were applied, and results were displayed as a median and interquartile range (IQR). When appropriate, Welch t-test was applied and mean with 95% confidential interval (CI) was displayed. Statistical power analysis of the various tests applied, as well as calculation of the corresponding sample size, were performed using G*Power v.3.1.9.6 for Mac (HHU, Düsseldorf, Germany). Since no pilot experiments have been performed previously, a posteriori power analysis was used.

The non-parametric Spearman’s rank correlation analysis was performed to find out the relationship of studied BMI, BP, and the study cytokines. In order to better estimate the size of the difference in the correlations of two pairs of variables, the variable delta r was used as an indicator, which was calculated by subtracting the smallest from the largest correlation coefficient according to the following formula: Δ*r* = *r*_max_ − *r*_min_. The higher the value of the coefficient (which is affected by both the density of the data spread and the direction of the correlations), the more pronounced and significant the differences between the compared pairs. Values below 0.4 are not considered a significant difference.

The *p* value of less than 0.05 (*p* < 0.05) was considered statistically significant for all used statistical tests.

An unsupervised machine-learning approach based on hierarchical cluster analysis with JMP Pro 16 software for Mac (SAS Institute, Cary, NC, USA) was used as a model to identify groups of studied factors that could be proposed as potential biomarkers for distinguishing between migraine patients and healthy controls.

All graphical images and statistical analyses were performed using GraphPad Prism 9.0 for MacOS software (GraphPad Software, San Diego, CA, USA).

## 3. Results

### 3.1. Association of BMI, BP, and Cytokines Regarding Migraine Type–Migraine with or without Aura

Correlation analysis reveals some significant associations regarding the type of migraine (without migraine, migraine with or without aura) of the studied cytokines, respectively, IL-8 (*r*_S_ = 0.38, *p* = 0.0036), PAI-1 (*r*_S_ = −0.43, *p* = 0.0018), and sICAM-1 (*r*_S_ = −0.37, *p* = 0.0072), show one positive and two negative associations (Figure 1A).

Migraine patients have higher plasma concentrations of IL-8 and lower concentrations of PAI-1 and sICAM-1 compared to healthy individuals (control), as shown in Figure 1B. However, with regard to the type of migraine, i.e., migraine with or without aura, IL-8 levels as well as PAI-1 and sICAM-1 do not differ significantly between patient groups (Figure 1C).

### 3.2. Correlations between BMI and BP in the Study Groups

We find significant correlation between BMI and systolic BP (*r*_S_ = 0.5793, *p* = 0.0380, Figure 2A) and also diastolic BP (*r*_S_ = 0.6077, *p* = 0.0276, Figure 2B) only in migraine patients with aura, and these associations of BP and BMI between two migraine groups are more expressed regarding systolic BP (∆*r*_S_ = 0.5156, Figure 2A). 

### 3.3. Correlations between BMI and Cytokines Levels in the Study Groups

We also find significant inverse correlations between BMI and MMP-9 (*r*_S_ = −0.5532, *p* = 0.0499) and direct correlation between BMI and IFN-γ (*r*_S_ = 0.6455, *p* = 0.0172) only in migraine patients with aura (Figure 3). For PAI-1, it is similarly correlated with BMI in both healthy individuals and patients with aura, but the correlation is not significant in patients with migraine without aura. On the other hand, the significance of the correlation between BMI and sVCAM-1, which is expressed in healthy individuals, disappears in both groups of patients.

It should be noted that the relationship between MMP-9 and BMI could be one of the molecular features to distinguish patients with migraine without aura from migraine with aura, and is maintained by the different correlation direction and the significant differences in the slopes of the regression curves (*p* = 0.0321), as well as by the high value of the correlation difference index (∆*r*_S_ = 0.9430) (Figure 3).

### 3.4. Correlations between BP and Cytokines Levels in the Study Groups

Both systolic and diastolic BP are directly correlated with MCP-1 in migraineurs without aura (*r*_S_ = 0.5442, *p* = 0.0499 and *r*_S_ = 0.5837, *p* = 0.0362), but not in patients with aura. However, in comparison to the control group, where there is also a significant, but negative, association between both respective factors (*r*_S_ = −0.4381, *p* = 0.0285 and *r*_S_ = −0.4762, *p* = 0.0161), the difference between the corresponding correlations of the two groups (control and migraine) is very pronounced. This is indicated by the significant differences in the slopes of the regression lines (*p* = 0.0096 and *p* = 0.0042) and the high values of the comparative indicator of correlations (∆*r*_S_ = 0.9823 and ∆*r*_S_ = 1.0599) (Figure 4). Taking these results together, it can be said that as blood pressure (both systolic and diastolic) increases, the blood concentration of MSP-1 increases in migraineurs without aura, contrary to what is observed in control individuals and also in migraineurs with aura, who have a decrease in MCP-1 concentration.

Similar opposite effects of correlations are observed in the analysis of sICAM-1, where they are most pronounced for diastolic pressure, but are only observed in migraine with aura, in contrast to MCP-1. Thus, a significant positive association between sICAM and diastolic BP is observed in the control group (*r*_S_ = 0.4178, *p* = 0.0422), but in migraine patients with aura it is completely opposite, i.e., negative, *r*_S_ = −0.4241, *p* = 0, 0421, the slopes differ significantly (*p* = 0.0124), while the correlation difference index is also high (∆*r*_S_ = 0.8419). A similar tendency is observed regarding sICAM-1 and systolic BP. Although a more pronounced positive relationship between sVCAM-1 and blood pressure is observed in patients with migraine aura (*r*_S_ = 0.5356, *p* = 0.0484, for diastolic pressure), no significant differences are found compared to controls.

However, in all cases represented in Figure 4, one can see a fairly pronounced difference in correlations between migraine patients without and with aura (see respective correlation difference indexes). 

### 3.5. Covariances of the Investigated Cytokines in the Study Groups

We find positive (direct) correlation between IL-8 and PAI-1 in healthy subjects (*r*_S_ = 0.34), while it does turn to the opposite direction (indirect correlation) in migraine patients: without aura, as feasible association between these two cytokines is not observed more (*r*_S_ = 0.07), while with aura it becomes negative (*r*_S_ = −0.37) (Figure 5A–C). 

In comparison to healthy control group, only one positive association of PA-1 (with IL-8) is observed in patients with migraine without and with aura. Many correlative relationships between PA-1 and other cytokines are found: positive with MMP-9 (*r*_S_ = 0.77), TGF-α (*r*_S_ = 0.82), and MCP-1 (*r*_S_ = 0.33), and negative one with sICAM-1 (*r*_S_ = −0.72) in migraine patients without aura (Figure 5B). On the other hand, in migraine patients with aura (Figure 5C), positive but less pronounced association remains only with MMP-9 (*r*_S_ = 0.46) and TGF-α (*r*_S_ = 0.62) (Figure 5A–C). 

MMP-9 shows only one inverse correlation with sICAM-1 (*r*_S_ = −0.40) in healthy subjects, and this correlation remains in migraine patients without aura (*r*_S_ = −0.37). However, in these patients, there are three direct correlations between MMP-9 and PAI-1 (*r*_S_ = 0.77), TGF-α (*r*_S_ = 0.45), and MCP-1 (*r*_S_ = 0.48). The situation is different in migraine patients with aura, where there is one new inverse correlation between MMP-9 and IL-8 (*r*_S_ = −0.37) and one new direct correlation between MMP-9 and PAI-1 (*r*_S_ = 0.46), but a correlation with TGF-α is less pronounced (*r*_S_ = 0.45) than in patients without aura (Figure 5A–C).

sVCAM-1 presents only one inverse correlation and it is with MCP-1 in healthy individuals (*r*_S_ = −0.48); the correlation is stronger in migraine patients without aura (*r*_S_ = −0.64), but in these patients, inverse correlations also appear, with IL-8 (*r*_S_ = −0.62). The situation is different in migraine patients with aura, where there is only one single correlation between sVCAM-1 and IL-8 (*r*_S_ = −0.49), but it is weaker than in patients without aura (Figure 5A–C).

sICAM-1 shows only inverse correlations. Correlation between sICAM-1 and MMP-9 is similarly expressed in healthy individuals and in migraine patients without aura (*r*_S_ = −0.40 and *r*_S_ = −0.37), but these patients have three other inverse correlations: sICAM-1 correlates with PAI-1 (*r*_S_ = −0.72), with TGF-α (*r*_S_ = −0.45), and with MCP-1 (*r*_S_ = −0.39). In patients with aura, the situation is different, with only two inverse correlations, where sICAM-1 correlates with IL-8 (*r*_S_ = −0.46) and also with TGF-α (*r*_S_ = −0.30) (Figure 5A–C).

TGF-α shows two pronounced direct correlations with PAI-1 (*r*_S_ = 0.82), and with MMP-9 (*r*_S_ = 0.71), and one inverse correlation between TGF-α and sICAM-1 (*r*_S_ = −0.45) in migraine patients without aura. TGF-α correlates with PAI-1 (*r*_S_ = 0.62); with MMP-9 (*r*_S_ = 0.45), it is less pronounced in migraine patients with aura but remains, and TGF-α with sICAM-1 (*r*_S_ = −0.30) is also less pronounced (Figure 5A–C).

In contrary to control, as well as to the migraine aura group, the most active and pronounced co-variability between MCP-1 and studied cytokines is observed in the migraine patient group without aura (Figure 5B). There are positive associations with PAI-1 (*r*_S_ = 0.33) and MMP-9 (*r*_S_ = 0.48), and negative associations with sVCAM-1 (*r*_S_ = −0.64) and sICAM-1 (*r*_S_ = −0.39). Interestingly, there is no correlation between MPC-1 and other study cytokines in patients with aura (Figure 5C). 

Compared to the control group, the covariance level of the investigated cytokines is increased in patients with migraine (Figure 5A,B), while the common profile of the co-variability of these factors differs quite clearly between the two migraine groups both without aura and with aura (Figure 5B,C).

## 4. Discussion

### 4.1. Cytokine Levels in Blood Serum Including Migraine Patients with and without Aura

Chemokines are now considered to stimulate the activation of trigeminal nerves as pain mediators in neurovascular inflammation, including the release of nitric oxide and other vasoactive peptides [5]. A published study shows that IL-8 level increases during migraine attacks and is caused by CGRP activation [12], but in our study, we found that IL-8 levels are also increased in migraineurs during their interictal period, which is consistent with one other study [13]. We found no difference in IL-8 levels between migraine patients with and without aura.

Interestingly, serum levels of PAI-1 and also sICAM-1 are reduced in migraine patients regardless of the presence of aura. Our PAI-1 results are consistent with another study of PAI-1 in migraine patients without aura during headache-free periods [14], although our finding also declines in patients with aura. We found that serum levels of sICAM-1 are also reduced in migraine patients. A possible explanation for this could be that nitric oxide contributes to the down-regulation of sICAM-1 during a migraine attack [15], but our data refer to the interictal period.

The study patients have an aura period characterized by visual disturbances before the headache starts. How trigeminal activation causes migraine headaches remains unclear, but cortical spreading depolarization (CSD) is known to correlate with migraine aura, and CSD plays a role in headache via parenchymal neuro-inflammatory signaling [16]. It is possible that IL-8 also plays some role in this neuroinflammatory pathway. Although we do not find that IL-8 levels are higher in patients with aura, there is a pronounced correlation (inverse) between IL-8 and PAI-1 only in patients with aura. Endothelial, hemostatic, and hemorheological factors are now believed to be involved in the pathogenesis of migraine [14].

### 4.2. Correlates of BMI and BP, Including Cytokines, in Migraineurs with and without Aura

BMI is strongly correlated only with two cytokines in our study, that is, MMP-9 and IFN-γ, but only in patients with aura.

We find differences between patients with and without aura regarding correlations between blood pressure type and cytokines. Both systolic and diastolic BP are strongly and directly associated with MCP-1 only in patients without aura. On the other hand, there is a strong inverse correlation between diastolic BP and sICAM-1, but, interestingly, direct correlation with sVCAM-1. Only one significant correlation (direct) is present in patients without aura, which is between systolic BP and sVCAM-1.

### 4.3. Cytokine Intercorrelations in Migraineurs with and without Aura

In the study, we obtained several close positive and negative associations between the cytokines IL-8, PAI-1, MMP-9, sVCAM-1, sICAM-1, TGF-α, and MCP-1. The strength of these associations differs depending on whether migraineurs are with or without aura during the interictal period.

The most correlations between cytokines are in migraine patients only without aura, such as correlations between MPC-1 and PAI-1 (direct correlation), MMP-9 (direct), sICAM-1 (inverse), and sVCAM-1 (inverse). Also, other correlations between cytokines are only in migraine patients without aura: between PAI-1 and sICAM-1 (inverse); between MMP-9 and sICAM-1 (inverse); and between TGF-α and PAI-1 (direct), MMP-9 (direct), and sICAM-1 (inverse). Several other correlations between cytokines are more pronounced in migraine patients without aura than in patients with aura: between PAI-1 and MMP-9 (direct); between MMP-9 and PAI-1 (direct), and TGF-α (direct); as well as between sVCAM-1 and IL-8.

Migraine patients with aura have distinct correlations that are not present in patients without aura, and of particular note is the correlation between IL-8 and PAI-1 (inverse correlation). There are two other significant correlations only in migraine patients with aura: the correlation between MMP-9 and IL-8 (inverse) and between IL-8 and sICAM-1 (inverse). Also, of note, MCP-1 does not correlate with any other study cytokine in patients with aura.

### 4.4. Potential Biomarkers to Distinguish Migraine Patients from Healthy Individuals

Several studies highlight the role of dysregulated immune responses in the pathophysiology of migraines [17], however, the role of inflammatory mediators is not yet fully understood [13]. Performed hierarchical cluster analysis (Figure 6), based on an ML modelling approach, reveals a group of three cytokines, PAI-1, TGF-α, and MMP-9, which could be used as potential markers to distinguish migraine patients from healthy individuals or subgroups of migraine patients.

Hierarchical cluster analysis (Figure 6A–C) shows that three cytokines, PAI-1, MMP-9, and TGF-α, can form a response similar (subcluster) in both groups of patients with migraine (without aura and with aura) compared with healthy people without migraine. Therefore, they may have the potential (combined with other measures such as levels of association between other cytokines, as well as cytokines with BMI and BP) as biomarkers for distinguishing migraine patients from healthy individuals. The main limitation of our study is the small number of patients studied.

## 5. Conclusions

Migraine patients have elevated levels of IL-8, but decreased serum levels of PAI-1 and sICAM-1 during the interictal period, regardless of aura;Pronounced association of BMI with BP (both systolic and diastolic), and also with IFN-γ and MMP-9 is observed only in patients with aura;In patients with aura, only diastolic BP correlates with sICAM-1 (and inversely compared to control) and sVCAM-1, but in patients without aura, both systolic and diastolic BP correlates with MCP-1, and these associations are opposite compared to the control;Regarding cytokine intercorrelations, there are three correlations in migraine patients with aura that are absent in patients without aura: between IL-8 and PAI-1; MMP-9 and IL-8; and IL-8 and sICAM-1;Migraine patients without aura, on the other hand, have correlations that patients with aura do not: between PAI-1 and MCP-1, sICAM-1; between MMP-9 and sICAM-1, MCP-1; between TGF-α and PAI-1, MMP-9, and sICAM-1; between sICAM-1 and MMP-9, PAI-1, and MCP-1; as well as between sVCAM-1 and MCP-1;There are also correlations that are present in all migraine patients, but are more pronounced in patients without aura: between PAI-1 and MMP-9, TGF-α; between MMP-9 and PAI-1, TGF-α; between TGF-α and PAI-1, MMP-9, and sICAM-1; as well as between sVCAM-1 and IL-8;PAI-1, TGF, and MMP-9 could be used as potential biomarkers to distinguish migraine patients from healthy individuals.

## Figures and Tables

**Figure 1 jcm-11-05696-f001:**
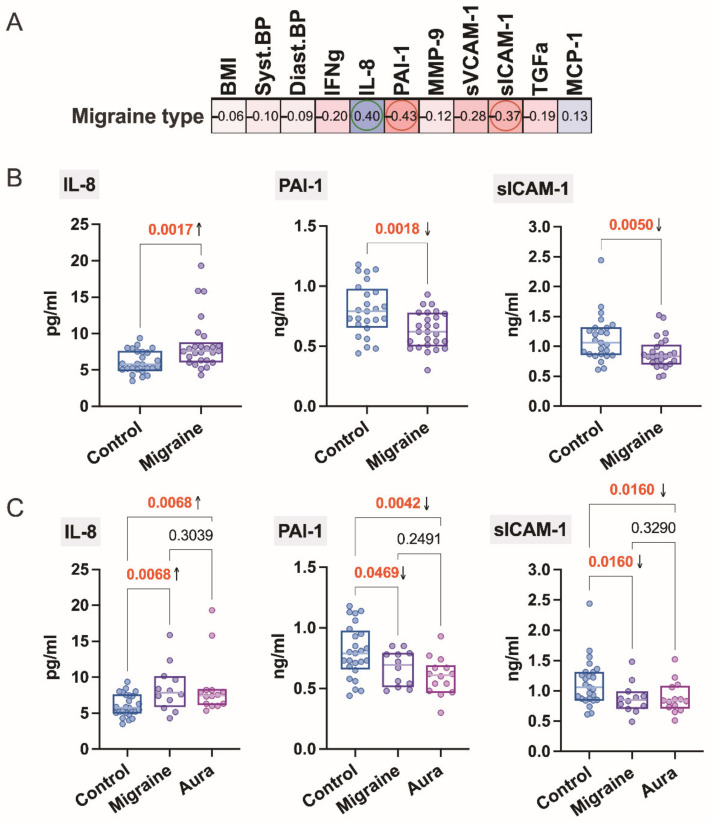
(**A**) Association of BMI, BP, and cytokines regarding migraine type (healthy individuals: control; patients with migraine without aura: migraine and with aura: aura); numbers represent respective Spearman’s rank correlation coefficient. (**B**) Comparison of the concentration of IL-8, PAI-1, and sICAM-1 in the blood of healthy individuals and migraine patients (migraine + aura), Mann–Whitney test. (**C**) Regarding studied types of migraine; Kruskal–Wallis’ test followed by two-stage step-up method of Benjamini, Krieger, and Yekutieli as post-hoc procedure for the adjustments of the significance level for multiple comparisons.

**Figure 2 jcm-11-05696-f002:**
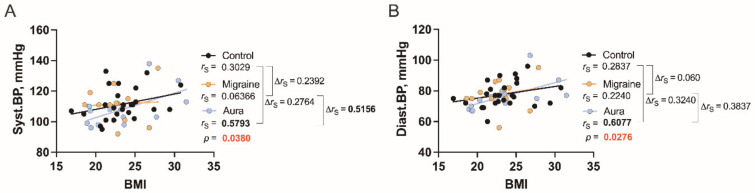
The relationship between BMI and BP ((**A**) systolic BP, (**B**) diastolic BP) in healthy individuals (control) and in patients with migraine without aura (migraine) and with aura (aura). *r*_S_: Spearman’s rank correlation coefficient, ∆*r*_S_: difference of coefficients.

**Figure 3 jcm-11-05696-f003:**
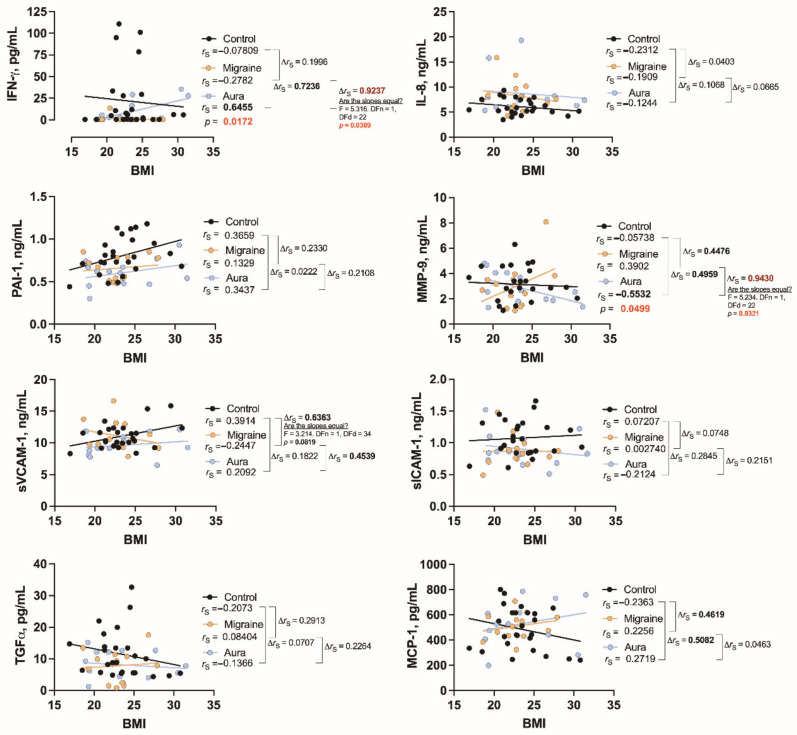
The relationship between BMI and cytokines studied in healthy individuals (control) and in patients with migraine without aura (migraine) and with aura (aura). *r*_S_: Spearman’s rank correlation coefficient, ∆*r*_S_: the difference between the correlation coefficients as an indicator of the magnitude of the differences (more significant differences are marked in bold).

**Figure 4 jcm-11-05696-f004:**
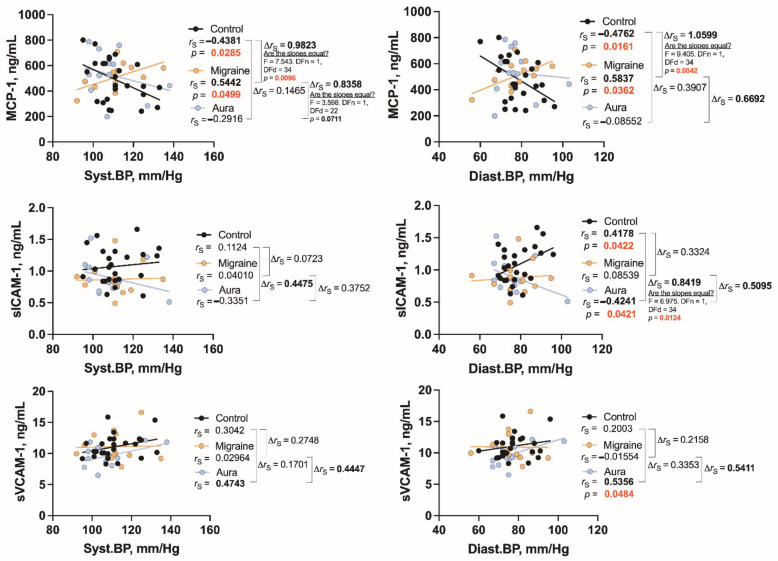
The relationship between BP (Syst.: systolic, Diast.: diastolic) and MCP-1, sICAM-1, and sVCAM-1 in healthy individuals (control) and in patients with migraine without aura (migraine) and with aura (aura). *r*_S_: Spearman’s rank correlation coefficient, ∆*r*_S_: the difference between the correlation coefficients as an indicator of the magnitude of the differences (more pronounced or significant differences are marked in bold).

**Figure 5 jcm-11-05696-f005:**
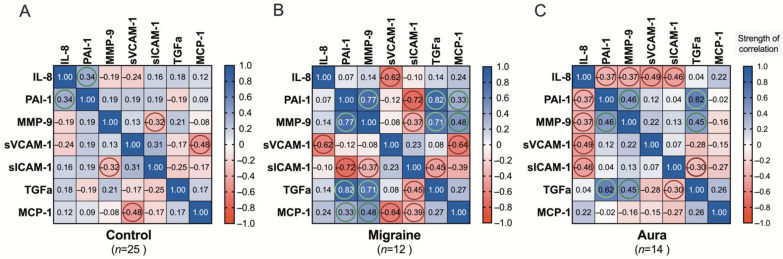
Spearman’s rank correlation (*r*_S_) matrix showing the interrelationships of the studied cytokines: (**A**) associations in healthy individuals (control); (**B**) associations in migraine patients without aura; (**C**) associations in migraine patients with aura. More pronounced associations are marked with colour circles.

**Figure 6 jcm-11-05696-f006:**
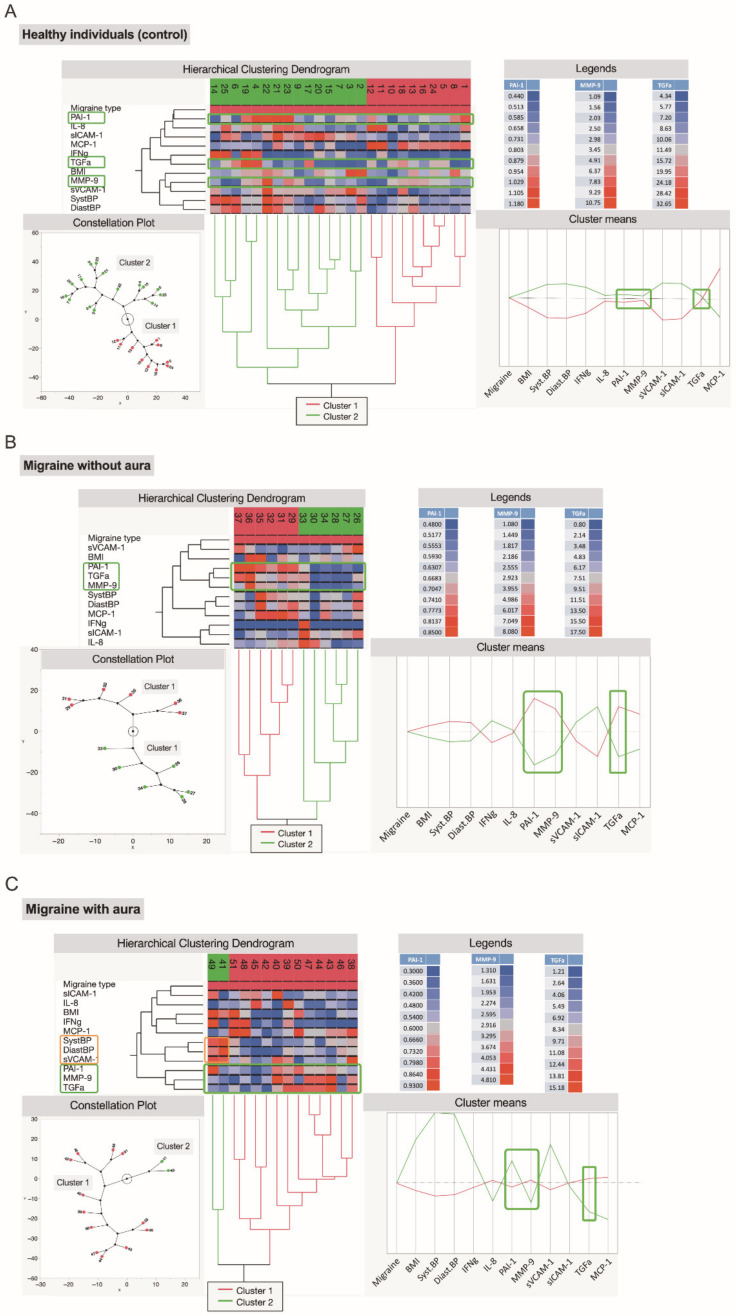
Cluster analysis of the studied groups: (**A**) healthy individuals (control), (**B**) patients with migraine without aura, (**C**) patients with migraine with aura. Results represented as hierarchical clustering dendrogram with constellation plot and cluster means. Green rectangles in this model mark three cytokines (PAI-1, TGFα, MMP-9) that could be used as putative markers for distinguishing migraine patients from healthy controls.

**Table 1 jcm-11-05696-t001:** The baseline characteristics.

	Control Group *n* = 25	Migraine Patients’ Group without Aura *n* = 12	Migraine Patients’ Group with Aura *n* = 14	*p* Value (Ordinary One-Way ANOVA)
Age, years	34.6 (1.3)	39.7 (1.5)	34.5 (2.3)	0.08
BMI, kg/m^2^	23.4 (0.6)	23.2 (0.8)	22.8 (1.0)	0.72
Systolic blood pressure, mm Hg	112 (2)	112 (3)	108 (3)	0.43
Diastolic blood pressure, mm Hg	78 (29)	78 (3)	77 (3)	0.59

The data are expressed as a number (*n*), or mean (±SD).

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
