# Peer review of "Association of Body Mass Index, Blood Pressure, and Interictal Serum Levels of Cytokines in Migraine with and without Aura"

_jcm, 2022, doi:10.3390/jcm11195696_

Round 1

Reviewer 1 Report

The authors investigated the BMI, BP, and other cytokines among migrain sufferes and heatlhy controls. They found that some cytokines are higher in migraine patients compared to the heatlhy controls.

Introduction

The content is fine, but the paragraphs seem to start abruptly. so please consider mentioning BMI, BP, and cytokine first and then explain each of them in each paragraph?

Methods

Line 44 Both groups were matched -> How did you matched them and confirmed statistically? If you select patients, please provide the propensity score. If they are subsequent series, please describe so.

Line 52 provide ethical approval number.

Line 58 Provide ELISA kit product number and company.

Statistics; Power analysis should be done.

Results;

Use box plot for non-normal distribution. Graph and bar for normal distribution.

Figure 1 Please provide actual pvalue for n.s. in the text. n.s. for figure is good, but provide actual numerical values in the text.

Figure 2 and Figure 3 and Figure 4 Please priovide p value for spearmans's r. Or, please describe in the method that r with bold or marked means with p < 0.05.

Discussion

Although the study patients had normal body weight and blood pressure, we found a significant association between BMI and 187 blood pressure (both systolic and diastolic) -> is this needed?

The results show that many cytokines and proteins differ between migraine and non-migraine patients. It is true. Why don't you start by summarizing the facts from this study in the first paragraph?

Maybe add subheadings to the Discussion section to make it a little easier to grasp the big picture.

Please mention a little bit about MEDICATION OVER USE HEADACHE as well as migraine. Early detection is necessary, and you have given us a valid biomarker.
PMID: 35043356

Author Response

Dear Reviewer,

Thank you very much for your thorough review. In the appendix we provide answers to your criticisms and suggestions.

  1. “Introduction. The content is fine, but the paragraphs seem to start abruptly. so please consider mentioning BMI, BP, and cytokine first and then explain each of them in each paragraph?”
  • We have started the second paragraph with the following sentence: “Overweight, and obesity in particular, is associated with a higher risk of developing not only diabetes, hypertension, cardiovascular disease and other diseases, but also a variety of pain disorders.” (lines of text: 31-32); but the third paragraph begins with the following sentence: “In general, hypertension plays one of the main roles in cardiometabolic diseases.” (35). The role of cytokines is briefly explained in the first part. Our proposal is not to expand it.

  1. “Methods. Line 44 Both groups were matched -> How did you matched them and confirmed statistically? If you select patients, please provide the propensity score. If they are subsequent series, please describe so.”
  • Because the initial number of patients that could be included in the study was not large, we selected patients according to a standard approach with exclusion criteria (this is described in the Materials and Methods section, see lines 44-51) to obtain groups with matched main characteristics - age, BMI, systolic and diastolic blood pressure (see Table 1).

  1. “Methods. Line 52 provide ethical approval number.”
  • Ethic Committee Name: Medical Ethics Committee of the Riga Stradins University. University Approval Code: 24.02.2012./1 (Line 55).

  1. “ Provide ELISA kit product number and company.”
  • Our manuscript now lists the manufacturers and catalogue numbers of all kits (lines of text: 63-67): Interleukin-8 (IL-8) – Merck Millipore, EZHIL8-100K; soluble intercellular adhesion molecule-1 (sICAM-1) – Merck Millipore, ECM335; soluble vascular cell adhesion molecule-1 (sVCAM-1) – Merck Millipore, ECM340; matrix metalloproteinase-9  (MMP-9) – Sigma-Aldrich, RAB0372-1KT; interferon gamma (IFN-γ) – Sigma-Aldrich, RAB0222-1KT; monocyte chemoattractant protein-1 (MCP-1) – Merck Millipore, EZMCP1-99KRM; transforming growth factor alpha (TGF-α) – Sigma-Aldrich, RAB0459-1KT were measured in plasma by ELISA method using TECAN Infinite 200 PRO multimode reader (Tecan Group, Ltd., Mannedorf, Switzerland).

  1. “Statistics; Power analysis should be done.”

  • Power analysis is the name given to the process for determining the sample size for a research study.  The technical definition of power is that it is the probability of detecting a “true” effect when it exists. Since no pilot experiments have been performed previously, a posteriori power analysis was used, but we didn’t include the results. We have added the sentence about power analysis in methodological part (lines of text: 76-707).
  • However, since most research studies are based on a random sample of the population of interest, power analysis results are meaningless because the random component in the study disappears after data is collected. Power analysis in general shows the likelihood that a statistical test or model will detect, say, estimated differences between two populations, such as a t-statistic to compare, say, mean blood pressure levels between two groups in a sample of interest in a prospective study. Once a sample is chosen, the results are no longer random and power analysis becomes meaningless for that particular study sample.

  1. “Results. Use box plot for non-normal distribution. Graph and bar for normal distribution.”
  • The bars in Figure 1 have been changed to boxplots (median with IQR).

  1. “Results. Figure 1 Please provide actual pvalue for n.s. in the text. n.s. for figure is good, but provide actual numerical values in the text.”
  • P-values are now provided in both Figure and text.

  1. “Results. Figure 2 and Figure 3 and Figure 4 Please provide p value for spearmans's r. Or, please describe in the method that r with bold or marked means with p < 0.05.”
  • P-values are now provided in both Figures and text.

  1. “Discussion. Although the study patients had normal body weight and blood pressure, we found a significant association between BMI and 187 blood pressure (both systolic and diastolic) -> is this needed?”
  • This sentence is deleted.

  1. “Discussion. The results show that many cytokines and proteins differ between migraine and non-migraine patients. It is true. Why don't you start by summarizing the facts from this study in the first paragraph?”
  • We added the subheadings suggested in the following question. Thus, the discussion is better structured in terms of content.

  1. “Discussion. Maybe add subheadings to the Discussion section to make it a little easier to grasp the big picture.”
  • We added the following subheadings: 1. Cytokine levels in blood serum including migraine patients with and without aura (209). The text of the paragraph, lines 211-216, is transferred to this subsection after line 219. 4.2. Correlates of BMI and BP, including cytokines, in migraineurs with and without aura (227). 4.3. Cytokine intercorrelations in migraineurs with and without aura (234). 4.4. Potential biomarkers to distinguish migraine patients from healthy individuals (249).

  1. “Please mention a little bit about MEDICATION OVER USE HEADACHE as well as migraine. Early detection is necessary, and you have given us a valid biomarker.”
  • Medication overuse headache (MOH) is associated with chronic migraine, but the patients in our study had episodic migraine.

Reviewer 2 Report

Comments for Association of Body Mass Index, Blood Pressure and Interictal Serum Cytokine Levels in Migraine with and without Aura

-       The introduction states that IL-1ß, TNF-a, IL-10 and IL-4 may play a role in pain modulation, however IL-8, IFN-g, TGF-a and MCP-1 were chosen. What is the incitement for this choice instead looking into cytokines with a potential involvement in migraine? Also, are there any prior evidence that these cytokines should be involved in hypotension or obesity?

-       Regarding the method, it is written that venous samples were taken in the interictal phase. Please elaborate on specific site of blood sampling. Furthermore, when was the samples taken relative to the participants’ last migraine attack.

-       Please provide the mean migraine and headache frequencies for the migraine patients – do you expect that the number of migraine/headache days might influence the serum levels? Seeing as a previous study have found differences in IL-6 and TNF-a interictally in migraine patient with episodic migraine compared to chronic migraine (e.g., Togha M, Razeghi Jahromi S, Ghorbani Z, et al. Evaluation of inflammatory state in migraineurs: a case-control study. Iran J Allergy Asthma Immunol 2020; 19: 83–90.)

-       From a statistical point of view, was adjustments for multiple comparison used?

-       Please add to the paragraph with statistical analysis that Spearman’s rank correlation was used for correlation analysis.

-       Result section: please provide p-values for the spearman correlation coefficients. Also on page 4, it says Correlation analysis results showed that migraine type (migraine with or without aura) and IL-8, PAI-1 and sICAM-1 had only 89 three stronger associations (rS = 0.38, rS = −0.43, rS = −0.37) . From a statistical point, this should be 2 weak association (r between 0.20-0.39) and one moderate association (r between 0.40-0.59).

-       Generally, for the discussion section, multiple findings are presented however the cause of the potential reason for these findings are lacking. An example is the correlations found only in migraine without aura patients – what could potentially explain the different correlations that were not seen in migraine with aura patients?

-       Regarding IL-8, the authors state that IL-8 increases during migraine attacks (Sarchielli et al., 2004), but the present study found that IL-8 levels are also increased during their interictal period, which is consistent with one other study (Duarte et al., 2015). Sarchielli et al. (2004) reported an increase ictally in jugular venous blood but not in peripheral venous blood samples during the attack. Outside attacks, IL-8 did not differ compared to controls in the same study. Meanwhile, Oliveira et al (2017) has shown a decrease in IL-8 compared to controls. What are the authors’ thoughts on these contradicting findings? For a quick overview of previous findings refer to a recent published systematic review written by Thuraiaiyah et al. (2022).

-       It is written that BMI was strongly correlated with MM-9 and IFN-g but only in patients with aura. What is the authors’ explanation for this finding? Is increased BMI caused by to the presence of aura, or could it be that patients with aura genetically are disposed to higher BMI and increased production of MM-9 and IFN-g?

Author Response

Dear Reviewer,

Thank you very much for your thorough review. In the appendix we provide answers to your criticisms and suggestions.

  1. “The introduction states that IL-1ß, TNF-a, IL-10 and IL-4 may play a role in pain modulation, however IL-8, IFN-g, TGF-a and MCP-1 were chosen. What is the incitement for this choice instead looking into cytokines with a potential involvement in migraine? Also, are there any prior evidence that these cytokines should be involved in hypotension or obesity?”
  • The PubMed database [https://www.ncbi.nlm.nih.gov/] contains articles (more or less) on all of the following cytokines: IL-8, sICAM-1, sVCAM-1, IFN-γ, PAI-1, MCP-1, MMP-9, TGF-α, which are included in our study. Our aim was to find out which cytokines might be more important in migraine. Studies show that practically all the mentioned cytokines are more or less associated with hypertension and obesity (we excluded such patients from the study).

  1. “Regarding the method, it is written that venous samples were taken in the interictal phase. Please elaborate on specific site of blood sampling. Furthermore, when was the samples taken relative to the participants’ last migraine attack”
  • Blood samples were taken from all patients in an outpatient setting within 10-14 days of the migraine attack (ictal phase).

  1. “Please provide the mean migraine and headache frequencies for the migraine patients – do you expect that the number of migraine/headache days might influence the serum levels? Seeing as a previous study have found differences in IL-6 and TNF-a interictally in migraine patient with episodic migraine compared to chronic migraine (e.g., Togha M, Razeghi Jahromi S, Ghorbani Z, et al. Evaluation of inflammatory state in migraineurs: a case-control study. Iran J Allergy Asthma Immunol 2020; 19: 83–90.)”
  • The frequency of migraine and headache in migraine patients was once a month, but migraine attacks lasted 48-72 hours. It is highly likely that the number of migraine/headache days may affect serum cytokine levels. It is possible that longer migraine attacks are associated with higher IL-8 levels in the interictal period.

  1. “From a statistical point of view, was adjustments for multiple comparison used?”
  • Adjusting for multiple comparisons means adjusting the level of significance to be more stringent in light of the increased experimentwise error rates. There are different standard procedures (post-hoc tests) for the adjustments of the significance level for multiple comparisons. For parametric analysis of variance, the Bonferroni, HSD Tukey or Scheffe tests and others are widely used as a posteriori procedure. We performed tests for normality, and after it was found that our data was not normally distributed, a non-parametric Kruskal-Wallis H test was applied with the respective post-hoc procedure recommended by the GraphPad Prism biostatistical software that we used for analysis (see the figure in the attachement and "Materials and methods" of the manuscript).

  1. “Please add to the paragraph with statistical analysis that Spearman’s rank correlation was used for correlation analysis.”

  • The statistical analysis section has been supplemented.

  1. “Result section: please provide p-values for the spearman correlation coefficients. Also on page 4, it says Correlation analysis results showed that migraine type (migraine with or without aura) and IL-8, PAI-1 and sICAM-1 had only 89 three stronger associations (rS = 0.38, rS = −0.43, rS = −0.37). From a statistical point, this should be 2 weak association (r between 0.20-0.39) and one moderate association (r between 0.40-0.59).”
  • P-values are now provided in both Figures and text.
  • The statement about "strong associations" is changed to "significant associations" and is based on the appropriate p-values (these are added to both figures and text).

  1. “Generally, for the discussion section, multiple findings are presented however the cause of the potential reason for these findings are lacking. An example is the correlations found only in migraine without aura patients – what could potentially explain the different correlations that were not seen in migraine with aura patients?”
  • Our findings suggest that cytokine intercorrelations differ between migraineurs with and without aura. At the moment, we cannot explain the obtained results with a deeper justification at the level of pathogenesis (referring to scientific articles), and it would be too much speculation.

  1. “Regarding IL-8, the authors state that IL-8 increases during migraine attacks (Sarchielli et al., 2004), but the present study found that IL-8 levels are also increased during their interictal period, which is consistent with one other study (Duarte et al., 2015). Sarchielli et al. (2004) reported an increase ictally in jugular venous blood but not in peripheral venous blood samples during the attack. Outside attacks, IL-8 did not differ compared to controls in the same study. Meanwhile, Oliveira et al (2017) has shown a decrease in IL-8 compared to controls. What are the authors’ thoughts on these contradicting findings? For a quick overview of previous findings refer to a recent published systematic review written by Thuraiaiyah et al. (2022).”
  • Yes, there are conflicting data on IL-8 in different studies, so we also wanted to obtain data from migraine patients directly in the interictal period. Our conclusion is that IL-8 plasma levels are also increased during the interictal period independently of the aura. Yes, it would be interesting to know what IL-8 levels could be in our patients during a migraine attack. Another study should be devoted to clarify this aspect.
  • It is possible that increased levels of IL-8 in the interictal period could parallel increased levels of IL-6 and TNF-a (these cytokines were not detected in our patients) and could contribute to the migraine chronification. This assumption follows to some extent from a recently published systematic review by Thuraiaiyah et al. (2022).

  1. “It is written that BMI was strongly correlated with MMP-9 and IFN-g but only in patients with aura. What is the authors’ explanation for this finding? Is increased BMI caused by to the presence of aura, or could it be that patients with aura genetically are disposed to higher BMI and increased production of MMP-9 and IFN-g?”
  • This question is difficult to answer qualitatively due to the lack of publications on this topic in the PubMed database.

Round 2

Reviewer 2 Report

No additional comments.